# The genetic basis of a social polymorphism in halictid bees

Sarah D. Kocher[1,2], Ricardo Mallarino [2,3,7], Benjamin E. R. Rubin[1], Douglas W. Yu[4,5,6], Hopi E. Hoekstra [2,3] & Naomi E. Pierce [2]

The emergence of eusociality represents a major evolutionary transition from solitary to group reproduction. The most commonly studied eusocial species, honey bees and ants, represent the behavioral extremes of social evolution but lack close relatives that are non-social. Unlike these species, the halictid bee *Lasioglossum albipes* produces both solitary and eusocial nests and this intraspecific variation has a genetic basis. Here, we identify genetic variants associated with this polymorphism, including one located in the intron of *syntaxin 1a* (*syx1a*), a gene that mediates synaptic vesicle release. We show that this variant can alter gene expression in a pattern consistent with differences between social and solitary bees. Surprisingly, *syx1a* and several other genes associated with sociality in *L. albipes* have also been implicated in autism spectrum disorder in humans. Thus, genes underlying behavioral variation in *L. albipes* may also shape social behaviors across a wide range of taxa, including humans.

[1] Department of Ecology and Evolutionary Biology, Lewis-Sigler Institute for Integrative Genomics, Princeton University, Princeton, NJ 08544, USA. [2] Department of Organismic and Evolutionary Biology, Museum of Comparative Zoology, Harvard University, 28 Oxford St, Cambridge, MA 02138, USA. [3] Department of Molecular and Cellular Biology, Howard Hughes Medical Institute, Harvard University, 52 Oxford St, Cambridge, MA 01238, USA. [4] Kunming Institute for Zoology, 32 Jiaochang Donglu, Kunming, Yunnan 650223, China. [5] Center for Excellence in Animal Evolution and Genetics, Chinese Academy of Sciences, 32 Jiaochang Donglu, Kunming, Yunnan 650223, China. [6] School of Biological Sciences, University of East Anglia, Norwich Research Park, Norwich NR4 7TJ, UK. [7] Present address: Department of Molecular Biology, Princeton University, Princeton, NJ 08544, USA. These authors contributed equally: Hopi E. Hoekstra, Naomi E. Pierce  Correspondence and requests for materials should be addressed to S.D.K. (email: skocher@princeton.edu) or to N.E.P. (email: npierce@oeb.harvard.edu)

Systems in which closely related populations (or species) differ in their social organization are extremely rare but ideal for elucidating the genetic mechanisms responsible for the evolution of social behavior. Within one family of bees, the halictid or "sweat" bees (Halictidae), eusociality has arisen independently at least twice[1,2]. Even within species, several halictids are socially polymorphic and capable of producing either solitary or eusocial nests. One of these species, *Lasioglossum albipes*, varies in social behavior between different populations[3] (Fig. 1). Unlike other halictid species where variation in eusociality appears to be the result of individual behavioral plasticity[4], in *L. albipes* differences in social behavior primarily occur among populations and common-garden experiments[3] suggest there is a strong genetic component underlying this variation.

Here, we take advantage of the natural variation found within *L. albipes* and use whole-genome resequencing of individuals collected from three social and three solitary populations (n = 25 individuals/population; Supplementary Data 1) to identify genetic differences associated with this social polymorphism. Our top candidates include a variant located in the intron of *syntaxin 1a* (*syx1a*), a gene that mediates synaptic vesicle release and plays a crucial role in neurotransmission. We show that *syx1a* is more highly expressed in social versus solitary bees, and we use cell-based assays to demonstrate that that this intronic variant can act as an enhancer that drives differences in reporter gene expression. Interestingly, we find that *syx1a* and several other genes associated with social behavior in *L. albipes* also have been implicated in autism spectrum disorder in humans. Taken together, our results suggest that changes in gene regulation may contribute to the evolution of eusociality, and that the genes underlying behavioral variation in *L. albipes* often have highly conserved functions and may also shape behavior across a wide range of insects and mammals.

## Results

### There have been repeated shifts in social behavior in *L. albipes*.

Whole-genome resequencing data identified a total of 2,655,960 genetic variants among 143 individuals of *L. albipes* that passed our quality filters (see Methods; Supplementary Data 1). We estimated the nucleotide diversity per site (π) to be 0.002, and we did not find differences in genetic diversity among social forms (Supplementary Table 1; two-sample *t*-test; $t_4 = -0.961$, $p = 0.391$). Next, to determine the extent of linkage disequilibrium (LD), we used 50 kb sliding windows across each genomic scaffold. As in honey bees[5], LD decays rapidly with physical distance:

we find a ~50% reduction in $r^2$ within 250 bp (Supplementary Fig. 1), providing near SNP-level resolution in subsequent genome-wide association analyses. This pattern is consistent across all populations and likely reflects high levels of genetic variation found within the species (potentially maintained via extensive gene flow, panmictic mating, and/or by the high recombination rates typical of social Hymenoptera[6]).

Using these genomic data, we first asked if there was evidence of gene flow among behavioral forms, or if the social and solitary populations represent distinct genetic lineages. Specifically, we estimated the genetic structure among individuals from the six population samples. The mean $F_{ST}$ between social and solitary forms is 0.06 (mean permuted $F_{ST} = 0.0009 \pm 0.0019$; permutation test, $p < 0.001$), suggesting that there is, or recently has been, gene flow among behavioral groups. These estimates are similar to those from honey bee populations, where $F_{ST}$ ranges from 0.05–0.2 across different populations[5]. Next, we implemented a principal component analysis (PCA) using a set of LD-pruned SNPs. The results demonstrate that individuals largely cluster by population, but populations do not cluster by social behavior (Fig. 2a). A population tree mirrors these relationships, with social and solitary populations repeatedly clustering together rather than separating by social form (Fig. 2b). Taken together, these analyses indicate substantial gene flow and/or shared evolutionary history between behavioral forms within *L. albipes*. Furthermore, the relationships among social and solitary individuals suggest repeated shifts in social behavior within this species—perhaps as a result of local adaptation[3,7]. Ancestral state reconstructions for this group suggest that *L. albipes* descended from a social ancestor;[2] therefore, this pattern likely reflects losses of eusociality in this species.

### Multiple genomic regions are associated with this social polymorphism.

The observed lack of genetic structure between social and solitary forms of *L. albipes* coupled with the repeated shifts in social behavior allows for a population-genomic approach to identify genetic variants associated with behavioral variation. We used a genome-wide, mixed-model association test (GEMMA[8]) that explicitly models and accounts for genetic relatedness while testing for correlations between phenotype and genetic variants. First, no variant was consistently fixed among all social versus all solitary populations, suggesting that there is not a single, shared locus shaping variation in social behavior in this species but rather that the genetic underpinnings of this trait are complex. Concordantly, we found eight regions containing 194 SNPs

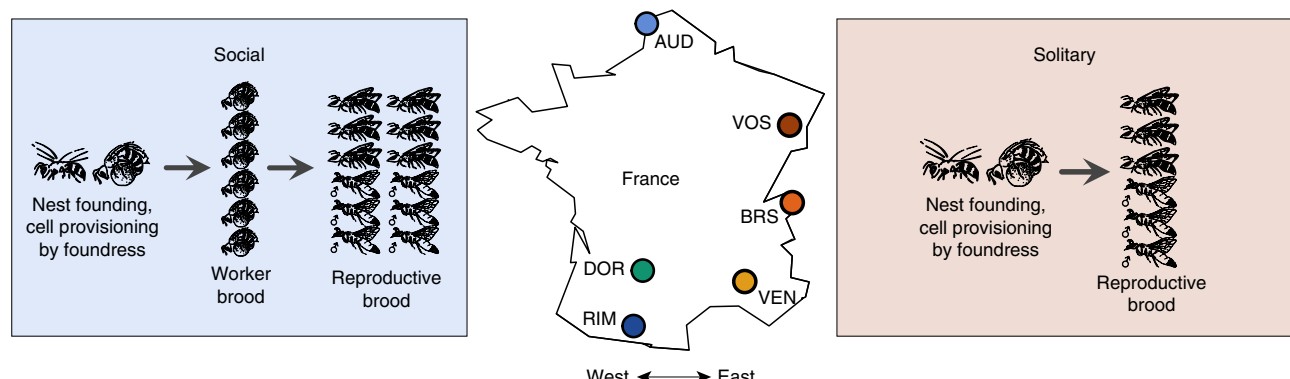

**Fig. 1** The socially polymorphic halictid bee *L. albipes*. Populations in western France are eusocial (blue and green colors), while populations in the east are solitary (red and orange colors). Eusocial females produce two broods: first workers, then reproductives, while solitary females produce a single, reproductive brood. This variation is strongly correlated with season length and mean temperature, with eusocial populations in regions with longer seasons and a warmer mean temperature than solitary populations[3]. Drawings by W. Tong

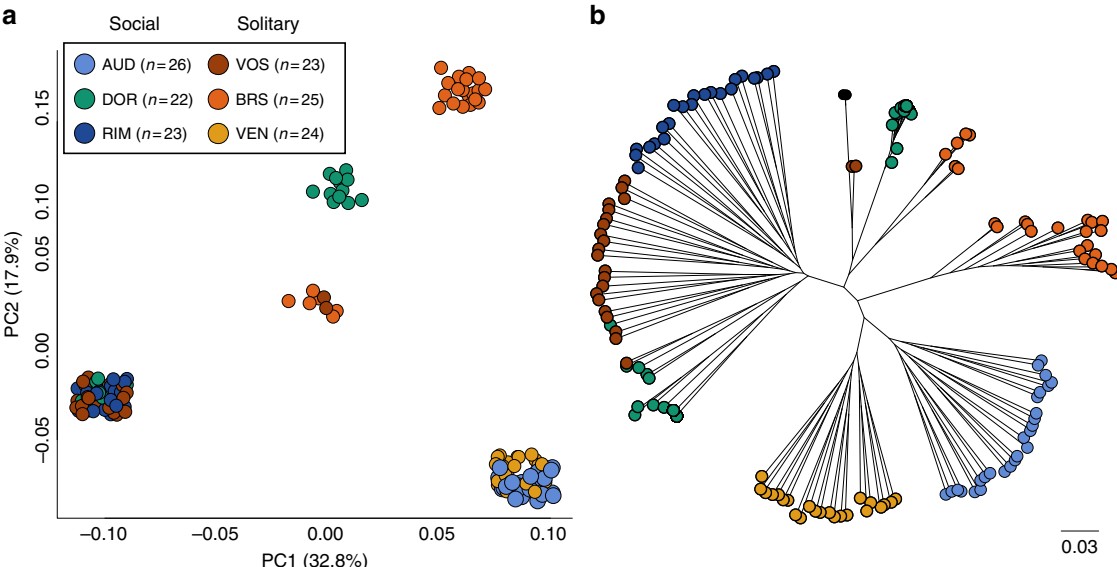

**Fig. 2** Repeated shifts in social behavior within *L. albipes*. **a** Genetic principal component analysis of six populations: three social (blues and greens) and three solitary (reds and oranges). **b** Population tree. Each dot represents an individual ($n = 143$). Analyses were run using LD-pruned SNPs ($n = 688,836$). Both the PCA and the population tree demonstrate that social and solitary forms are not incipient species; instead, the multiple groupings of social and solitary populations are consistent with repeated shifts in social behavior within *L. albipes*, likely the result of local adaptation

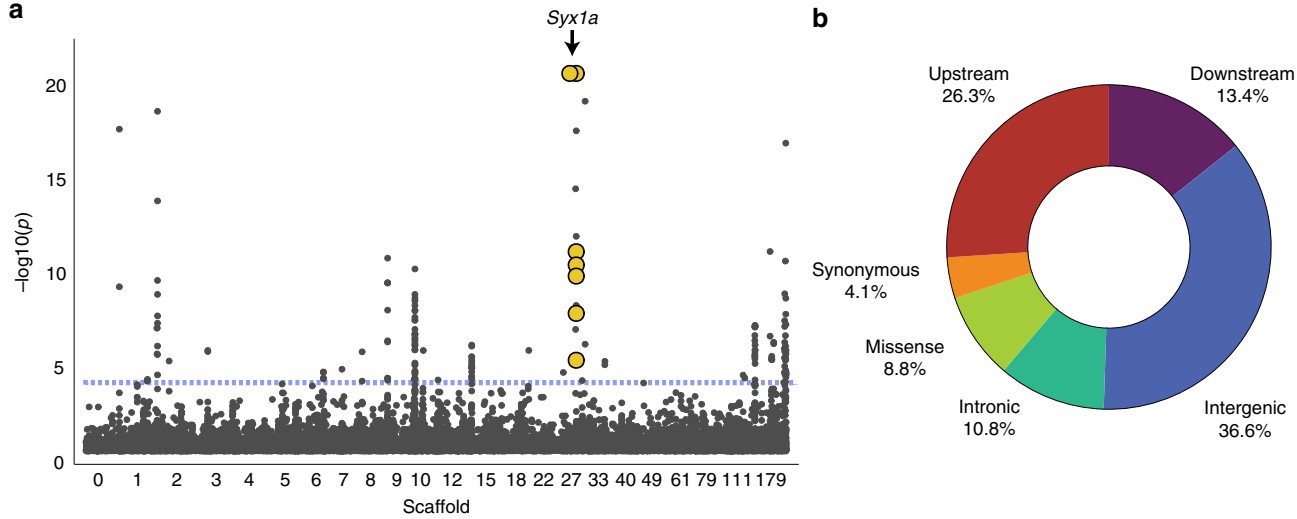

**Fig. 3** SNPs associated with social behavior and their genomic location. **a** Manhattan plot of the genome-wide associations ($n = 71$ social, 72 solitary bees). Each point represents a single SNP and its -log10 *p*-value (FDR-corrected for multiple testing). Only SNPs with FDR < 0.2 are included. SNPs associated with *syntaxin 1a* are highlighted (yellow). **b** Multiple SNPs are associated with variation in social behavior in *L. albipes* ($n = 194$ SNPs), ~40% of these occur in regulatory regions neighboring coding exons (i.e., $n = 45$ within 5 kb upstream of the transcription start site; $n = 28$ within 1 kb downstream of the last codon)

passing a genome-wide, FDR-corrected significance threshold[9] of $5 \times 10^{-5}$ (which roughly corresponds to a raw-*p* threshold of $5 \times 10^{-9}$; Fig. 3; Supplementary Data 2), suggesting that multiple regions throughout the genome contribute to intraspecific behavioral variation in *L. albipes*.

**Variants are enriched in both regulatory and coding regions.** The candidate SNPs fall within 10 kb of 62 genes, and many of these differences are located in potential regulatory regions[10]. In fact, 40% of identified SNPs fall nearby genes, either 5 kb upstream of the transcription start site ($n = 45$) or 1 kb downstream of the

last codon ($n = 32$), a significantly greater proportion of variants in these regions than expected by chance (hypergeometric test, *p* = $1.4 \times 10^{-5}$). Moreover, 17 of these 194 SNPs, located in nine different genes, are predicted to be non-synonymous variants (Supplementary Table 2; hypergeometric test, $p = 1.3 \times 10^{-11}$), and eight variants occur at synonymous sites (hypergeometric test, $p = 0.02$). Similar proportions of coding to non-coding changes were observed in a comparison of stickleback freshwater and marine morphs[11]. Taken together, these results suggest that changes in both gene regulation and coding sequence play an important role in the social polymorphism within this species.

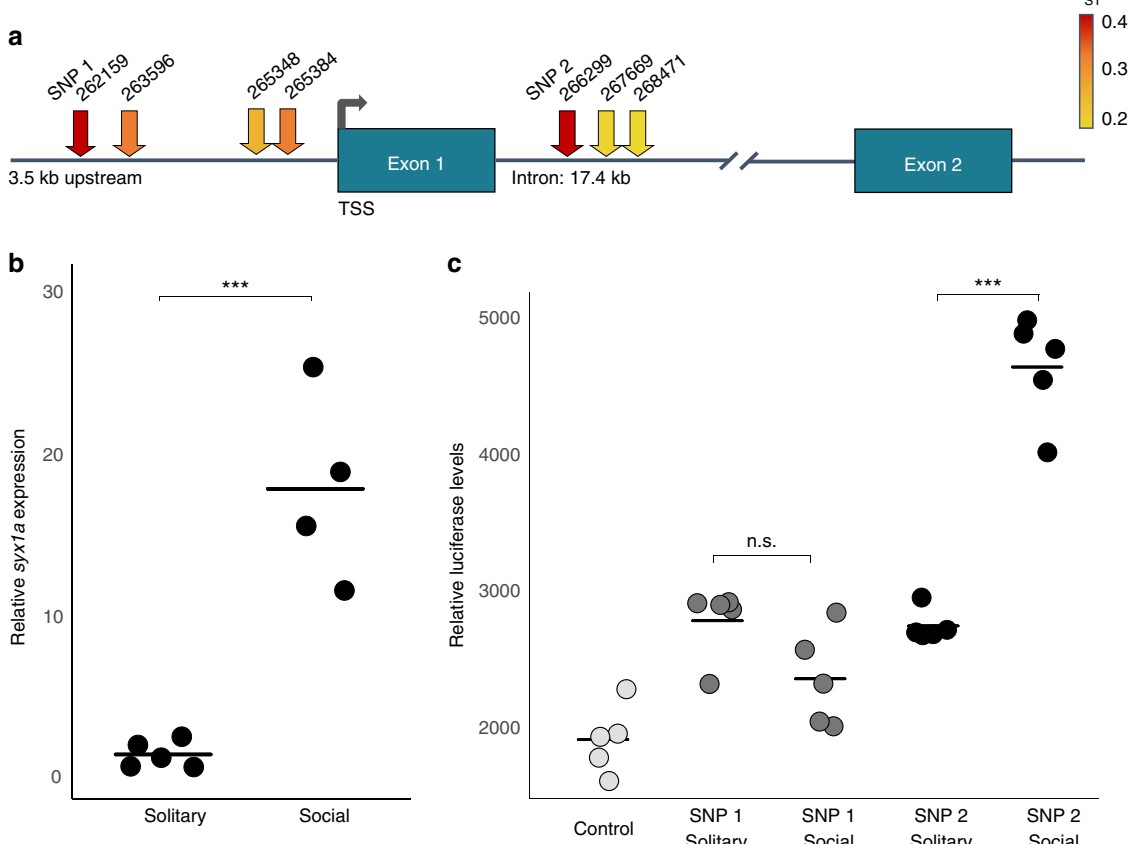

**Fig. 4** Functional effects of regulatory SNPs in the *syx1a* locus. **a** Schematic of *syx1a*. Arrows denote locations of the seven SNPs associated with social behavior in *L. albipes*. Arrow colors illustrate $F_{ST}$ between social forms at each SNP. Two SNPs, labeled SNP 1 and 2 (red), have the highest $F_{ST}$ estimates, and are in strong LD with each other ($r^2 = 0.803$), compared with the mean pairwise value among all seven SNPs ($r^2 = 0.18$). **b** Quantitative PCR identified differences in *syx1a* brain gene expression between social forms. Expression values are plotted as the fold change of normalized *syx1a* expression levels. Social individuals ($n = 4$) have significantly higher levels of *syx1a* brain gene expression than solitary individuals ($n = 5$; $t$-test, $t = -6.32$, $p = 0.0004$). **c** Luciferase reporter assays test if candidate SNPs in the regulatory regions of *syx1a* affect enhancer activity ($n = 5$ replicates per group). Both tested regions show enhancer activity relative to control (Tukey's honestly significant difference (HSD) post-hoc test, $p < 0.05$). There is no significant difference between social and solitary alleles of SNP 1 (Tukey's HSD, $p = 0.1773$), but the social allele of SNP 2 drives ~1.5 times higher reporter expression than does the solitary allele (Tukey's HSD, $p < 0.0001$), consistent with the qPCR assay (**b**)

In total, we identified 62 candidate genes that either contained coding changes or were associated with nearby regulatory variants (within 10 kb). Gene ontology analyses performed on these candidates identified a significant overrepresentation of loci associated with neurotransmission (hypergeometric tests; SNARE binding, $p = 0.001$; regulation of neurotransmitter secretion, $p = 0.02$) and metabolism (TOR signaling, $p = 0.02$; negative regulation of insulin-like growth factor receptor signaling pathway, $p = 0.004$), among others (Supplementary Data 3). These gene functions suggest a link to behavioral variation, but because behavioral variation is tightly linked to environmental conditions in this species, some may also reflect associations driven by differences in climatic conditions. Overall, the mechanisms linked to facultative social behavior in halictids appear similar to those associated with obligate eusocial behavior in other species[12].

**Highly conserved genes, including *syx1a*, shape social behavior in bees and humans**. Of the 194 SNPs associated with social behavior, 21 are clustered in or nearby six candidate genes implicated in human autism (Supplementary Table 3), a significant overrepresentation (hypergeometric test, $p = 0.001$). This result is consistent with the observation that genes whose expression level is correlated with social responsiveness in honey bee workers include genes implicated in human autism spectrum disorders (ASD)[13]. Thus, our results, which focus on genetic sequence variants rather than correlations with gene expression, independently support the idea, with data from a different species and a different social behavior, that a highly conserved set of genes or pathways may be involved in shaping social behavior across both insects and vertebrates, including humans[14].

**An intronic variant in *syx1a* reconstructs the expression differences between social forms**. Our top candidate SNPs fall within a single peak on scaffold 28 (Fig. 3a). This peak contains variants exhibiting the strongest association with social behavior and includes seven SNPs clustered in regulatory regions surrounding an ASD-associated gene, *syntaxin 1a* (*syx1a*; Fig. 4a; Supplementary Table 4). *Syx1a* is a key component of the SNAP/SNARE complex and is critical for binding synaptic vesicles for their subsequent fusion and release of neurotransmitters at chemical synapses[15]. Changes in *syx1a* expression levels have been associated with social behavior in other species. For example, in migratory locusts, an increase in *syx1a* expression is associated with the transition from solitary to gregarious life cycles[16]; in

honey bees, changes in *syx1a* protein expression are correlated with olfactory learning[17]; in mice, *syx1a* knockouts show unusual social behaviors and deficits in synaptic plasticity and memory formation[18]; and, in humans, *syx1a* has been repeatedly associated with ASD[19–21]. Thus, *syx1a* represents an exciting candidate for harboring causal mutations that contribute to the natural evolution of transitions in social behavior among halictid populations.

First, to test if solitary and social bees show differences in *syx1a* expression, we used quantitative, reverse transcription PCR and found that bees from social populations have significantly higher levels of *syx1a* brain gene expression than those from solitary populations (Fig. 4b; *t*-test, t = −6.32, *p* = 0.0004). Second, to test if any of these seven SNPs—all of which are within 3.5 kb upstream of the start site (*n* = 4) or located in the first intron (*n* = 3)—affect *syx1a* gene expression, we measured divergence between social and solitary forms. As expected, $F_{ST}$ between social forms is elevated at each of these seven SNPs, ranging from 0.16–0.38 (vs. a genome-wide $F_{ST}$ = 0.06).

Two SNPs exhibited exceptionally high levels of $F_{ST}$ (0.38 and 0.37, respectively) between social and solitary forms (Fig. 4a). These SNPs are in strong LD with each other ($r^2$ = 0.803) despite being separated by 4.14 kb (SNP 1; LALB_28:262159 and SNP 2; LALB_28:266299), compared with LD between the other five SNPs, which is similar to genome-wide patterns (Supplementary Fig. 2). Both of these SNPs are located in regions with predicted binding motifs[22]. Moreover, at both SNPs, the derived allele (by comparison with the outgroup genotype in *L. calceatum*)[2] is present at higher frequency in social populations.

To determine whether SNP 1 and/or SNP 2 can affect *syx1a* expression, we tested their effect on gene expression using cell-based assays. Specifically, we inserted ~800 bp of sequence containing the SNP of interest from either the allele at higher frequency in social populations (henceforth the "social allele") or the allele at higher frequency in solitary forms (the "solitary allele"; Supplementary Figs. 3–4; ~400 bp on either side of the SNP, no other SNPs associated with sociality were identified in these regions) in an enhancer reporter vector and tested the ability of these sequences to drive luciferase expression. While both regions act as mild enhancers in our assay, only the second variant (SNP 2) had a strong and highly significant difference in expression between the social and solitary alleles (Fig. 4c; one-way ANOVA, $F_{4,20}$ = 65.0119, *p* < 0.0001). Specifically, the social allele drives higher expression of the reporter gene than the solitary allele (Tukey HSD, *p* < 0.0001), similar to the degree and direction we observed in the expression patterns of *syx1a* alleles in wild-caught social and solitary individuals (Fig. 4c). Given that both SNPs are in strong LD, it is also possible that SNP 1, which is upstream of the start site, may act as a promoter (rather than an enhancer); thus, we tested both variants using a promoter reporter assay, but found no difference in reporter expression between the solitary and social alleles for either SNP (Supplementary Fig. 5). Together, these experiments demonstrate that SNP 2 has a clear functional effect on *syx1a* expression levels, and in the direction predicted by expression in wild-caught individuals in *L. albipes*. Future functional studies in *L. albipes* will help elucidate the precise neurobiological mechanisms by which *syx1a* and other associated genes shape variation in social behavior within this single species.

## Discussion

Because the shifts in social behavior in *L. albipes* are more likely to be the product of losses of eusociality rather than independent gains[2], many of the genetic associations with variation in social behavior identified in this study are likely to represent mechanisms contributing to the maintenance or disruption of eusociality rather than to its precise evolutionary origins. Nonetheless, here the derived allele of *syx1a*, which occurs in an extended haplotype block, is associated with eusocial populations, suggesting this allele may be contribute to the maintenance or elaboration of social behavior in this system.

The evolution of eusociality has fascinated biologists, including Darwin[23], for centuries, and the repeated gains and losses of social behavior found in halictid bees provide a powerful lens to study this key evolutionary innovation. The identification of several genetic changes associated with the social polymorphism in *L. albipes* suggests that multiple loci underlie the evolution and maintenance of social complexity. Many of these variants occur in regulatory regions, including a functionally-relevant SNP located in an intron of *syntaxin 1a*, a gene that regulates neurotransmitter release and that has been implicated in human autism. Thus, our results are consistent with previous work suggesting that changes in gene regulation play key roles in the evolution of social behavior[24] and that genes known to be associated with social behavior in *L. albipes*, such as *syx1a*, may also shape social behavior in other, distantly-related species.

## Methods

**Experimental design**. The halictid bee species *L. albipes* varies in social behavior across Germany and France. Previous common-garden experiments have demonstrated that this variation has a strong genetic basis[3]. In this study, we use whole-genome resequencing from 150 individuals sampled in three social and three solitary populations (*n* = 25 individuals/population) across France to conduct a genome-wide association study and identify genetic variants associated with this intraspecific social polymorphism.

**Improved *L. albipes* genome assembly and annotation**. The draft *L. albipes* genome assembly consisted of 41,433 scaffolds with an N50 of ~616 kb, a total assembly size of 341 Mb, and an estimated total size of 416 Mb[25]. Here, we used DNA extracted from a single female collected in Le Brassus, Switzerland (solitary) to construct a sequencing library with 10X Genomics technology (10X Genomics, Pleasanton, CA). We then generated a de novo genome assembly using SuperNova software (version 1.0)[26]. This resulted in an improved genome assembly with 15,715 scaffolds, an N50 of 4.4 Mb, and a total assembly size of 288 Mb. To further improve our assembly, we used MeDuSa (v1.6)[27] and GapFiller (v1.10)[28]. Our final assembly had 3362 scaffolds, an N50 of 4.8 Mb, and total assembly size of 338 Mb. Gene annotations were transferred by reciprocal blasting the coding sequences of each genome annotation (draft and improved assembly) against each other and selecting the reciprocal best hit. Finally, the gene set was annotated using the trinotate pipeline. The previous assembly contained 13,448 annotated genes and a genome-wide BUSCO[29] score of 96.5%; the new assembly contains 15,905 predicted genes with a genome-wide BUSCO score of 95.6%.

**Sample collection and library construction**. We collected 25 haploid males from three social and three solitary populations of *L. albipes* (Fig. 1; Supplementary Data 1). The behaviors of females at each site had been previously established through field and lab work[3,7]. As an outgroup for downstream analyses, we also collected and sequenced the genomes (as described below) of two males of the sister species, *Lasioglossum calceatum*. These two individuals were collected at two of the *L. albipes* population sites (one *L. albipes* social site and one *L. albipes* solitary site; *L. calceatum* is eusocial at both of these sites[3,30]). All individuals were captured while foraging, immediately flash frozen in liquid nitrogen, and stored at −80 °C until further processing. We extracted DNA using Qiagen DNeasy kits (Qiagen, Valencia, CA, USA) with one modification from the standard protocol: samples were incubated in proteinase K overnight at 50 °C. Next, we sequenced ~900 bp of the cytochrome oxidase I gene (COI) from each individual to confirm species identification (see Supplementary Table 5 for primers). We then constructed paired-end 2 × 100 bp Illumina libraries using the Apollo 324 robot and Prep X ILM 32i kits (Wafergen, Freemont, CA, USA).

**Genome sequencing and variant calling**. We resequenced 150 *L. albipes* genomes and 2 *L. calceatum* individuals to ~20× coverage per individual (Supplementary Data 1) with 2 × 100 bp paired-end reads on an Illumina HiSeq 2000 with v3.0 reagents at BGI in Shenzhen, China. All individuals were individually indexed, then sets of six individuals were pooled and sequenced across 2 lanes, for a total of 51 sequencing lanes. Fastq files were deduplicated using FASTUNIQ[31]. We next mapped these raw sequences to the updated reference genome with BWA (bwa-mem)[32], and called SNPs using FreeBayes[33] using the default parameters. We excluded indels and variants with a quality score less than 30 from downstream

analyses. Because we initially used haploid males to generate the resequencing data, we also excluded variants with heterozygous calls in 15 or more individuals (~ 10% of the called variants), as were any individuals with high (> 30%) levels of heterozygosity ($n = 3$). Lastly, we excluded variants uncalled in $> = 16$ individuals, as well as sites with MAF < 0.05 or that were not biallelic. After filtration, we had 2,655,960 SNPs scored in 143 *L. albipes* from three social ($n = 71$) and three solitary ($n = 72$) populations and two outgroup *L. calceatum* individuals.

**Population-genetic analyses**. We calculated population-genetic parameters using VCFtools[34], PLINK[35], and EIGENSOFT;[36] we excluded *L. calceatum* from these population analyses. We estimated Watterson's theta from the number of segregating sites called within *L. albipes* after quality filtration ($n = 2,655,960$ SNPs) and with the total size of the genome assembly (338 Mb). LD was estimated in 50 kb windows across each scaffold using PLINK with the following options: --r2 --ld-window-r2 0 --ld-window-kb 50000 (Supplementary Fig. 1). We ran calculations on each population independently as well as on all populations pooled together; the two approaches produced similar results (Supplementary Table 1). We calculated whole-genome $F_{ST}$ statistics using EIGENSTRAT comparing all social to all solitary populations. To estimate the statistical significance of differentiation between social and solitary populations, social designations were randomly shuffled 1000 times for each individual, and $F_{ST}$ was estimated using a randomly-selected subset of 100,000 SNPs. Weir and Cockerham's per site $F_{ST}$ was calculated using VCFtools (v0.1.12b). Then, we conducted genome-wide demographic analyses using an LD-pruned dataset of 688,836 SNPs. We pruned SNPs in PLINK (v1.9) with the following options: --indep 50 5 2. Using EIGENSOFT, we conducted a genetic principal components analysis (Fig. 2a). We also generated a population tree using FastTree[37] (-gtr –nt), compiled with --DUSE_DOUBLE (to allow for more precise calculations on short branches) (Fig. 2b).

**Genome-wide associations**. To account for underlying population structure and relatedness among individuals within populations, we used GEMMA[8] to fit a univariate, linear mixed model to test for genotype–phenotype associations. This method includes a relatedness matrix as a covariate in the analysis to account for population stratification and sample structure. We classified each individual as either social or solitary based on its population of origin, and tested for an association between social behavior and the focal SNP. We then applied an FDR correction to account for multiple testing.

**Variant annotation**. We annotated all variants using SNPeff (v4.3i)[38] and conducted gene ontology analyses using the GOseq Bioconductor package[39]. We summarized redundant GO terms in REVIGO[40].

**Overlap with autism-associated genes in humans**. To assess overlap with autism-associated genes in humans, we used a curated set of genes associated with Autism Spectrum Disorder from the Simons Foundation Autism Research Initiative (SFARI). This gene list includes all known human genes associated with ASD, including genes identified in genome-wide association studies, genes where rare mutations have been linked to ASD, and genes linked to syndromic autism. We used the full list of genes included in this database. We matched *L. albipes* genes to human orthologs using of reciprocal best-blast hits between *L. albipes* and human, and also confirmed annotations with SwissProt homologies generated using the trinotate pipeline on the Lalb_v3 OGS[41].

**Selection of candidates for functional tests**. The strongest association between social behavior and genetic variation was localized to a region on scaffold LALB_28. The top variants in this region included seven SNPs located either just upstream (Supplementary Fig. 3) or in the first intron of *syntaxin 1a* (Supplementary Fig. 4). We elected to focus our functional analyses on two of these SNPs because: (1) they showed the highest estimates of divergence (of the seven SNPs in *syx1a*) between all social and all solitary populations, and (2) these two SNPs were in strong LD with each other (Supplementary Fig. 2). Because both SNPs are located in non-coding regions, we hypothesized that either one or both was likely to affect mRNA expression level of *syntaxin1a*.

**Quantitative real-time PCR**. We collected five social and five solitary individuals (males) from two sites: RIM (social) and BRS (solitary). We dissected brains in RNAlater (Qiagen) on ice and removed optic lobes to enrich for mushroom body tissue, a brain region associated with learning, memory, and sensory integration in arthropods[42] and with variation in social behavior in sweat bees[43]. We isolated RNA using the Picopure RNA Isolation kit (Thermo Fisher), following the recommended protocol, and quantified the RNA using the Qubit RNA HS Assay kit and a Qubit Fluorimeter. Following quantification, we used 100 ng of RNA for cDNA synthesis with SuperScript III Reverse Transcriptase (Thermo Fisher). We next designed primers for *syx1a* and two well-established housekeeping genes in bees: *Rps18* and *GAPDH*[44]. All primers span an exon junction to avoid amplification of DNA contaminants, and PCR products were run on a gel following amplification to confirm that only a single target was amplified. We then ran qPCR assays using SYBR green fluorescent dye on an ABI Bioanalyzer with 40

amplification cycles. We analyzed the resultant data by calculating the mean delta $C_T$ values for *syx1a* relative to *Rps18* for each biological replicate (each sample was run in triplicate), and then calculated relative expression levels as the fold change of normalized *syx1a* expression for each sample relative to the average delta CT value for solitary bees. We used a *t*-test to test for significant differences in mRNA expression among social forms. Analyses of *syx1a* expression levels relative to the second housekeeping gene *GAPDH* generated the same statistical outcomes. All individuals were Sanger sequenced to confirm genotypes (Supplementary Fig. 6).

**Luciferase assays**. To test whether or not our focal SNPs impacted gene expression levels, we cloned the candidate regulatory elements identified in the *syntaxin* locus. Specifically, we amplified an 866 bp region encompassing SNP 1 (forward primer: 5′-TTTGGGCCTGTGTGTTTGTA-3′; reverse primer: 5′-GCTACCAGAGGACGACGAAG-3′) and a 790 bp region encompassing SNP 2 (forward primer: 5′-TTGTTATGATTCCCCGTGGT-3′; reverse primer: 5′-CTGCCGGTACTCTCGTTCTC-3′). We used DNA from the same social and solitary individuals included in the resequencing dataset and verified the genotypes of each individual using Sanger sequencing. Then, we cloned these amplicons into the pLightSwitch Long-range Enhancer Reporter Vector or pLightSwitch Promoter Reporter Vector (Switchgear Genomics, Active Motif) and verified all constructs with Sanger Sequencing. We used FuGENE HD (Active Motif) to transfect *Drosophila* S2R+ cells with each construct. The day before transfection, we seeded cells at a density of $1 \times 10^4$ cells per well and transfected the different constructs using a FuGENE HD to plasmid ratio of 3:1 (300 nL FuGENE HD:100 ng plasmid DNA per well). After 48 h, we harvested and processed the cells using the LightSwitch Luciferase Assay Kit (Switchgear Genomics), following the protocol guidelines, and measured levels of luciferase activity using a SpectraMax L luminometer (Molecular Devices). We performed all luciferase experiments using five replicates per construct. We determined statistical significance using a one-way ANOVA and post-hoc comparisons were performed using a Tukey's HSD.

## Data availability

All the data are available from the authors upon request. Sequencing data has been deposited at the NCBI SRA, accession PRJNA413432.

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

## Acknowledgements

We thank C. Plateaux-Quenu and L. Plateaux for their invaluable guidance and input on this project. We thank J. Couget, S. Picard, K. Turner, and the BGI team for assistance with molecular work, and A. Brelsford, M. Chapuisat, A. Finkelstein, R. Jeanson, L. Keller, D. Michez, L. Murphy, M. Podolak, J. Purcell, L. Pellissier, P. Servigne, W. Tong, and E. Youngsteadt for assistance in the field. We also thank W. Tong for providing the drawings in Fig. 1. This work was supported by NSF-IOS 1257543 (to N.E.P. and H.E.H.) and by CAS- GYHZ1754, QYZDY-SSW-SMC024, MOST-2012FY110800, 20080A001, and the Kunming Institute of Zoology's State Key Laboratory of Genetic Resources and Evolution (to D.W.Y.). S.D.K. was supported by Foundational Questions in Evolutionary Biology and National Institute of Food and Agriculture postdoctoral fellowships, grants from the Putnam Expeditionary Fund of the Harvard Museum of Comparative Zoology, and Princeton University. R.M. was supported by the Howard Hughes Medical Institute (to H.E.H.). H.E.H. is an Investigator of the Howard Hughes Medical Institute.

## Author contributions

S.D.K. conceived of the project; N.E.P. and H.E.H. helped design the approach, secure funding, and supervise the research throughout. D.W.Y. helped provide funding for the genome sequencing and actively participated in the implementation and interpretation. S.D.K. collected the samples, generated and analyzed the population-genetic data, and drafted the initial paper. R.M. designed and conducted the luciferase assays. B.E.R.R. generated the updated *L. albipes* genome assembly. All authors discussed the results and their implications, provided comments and revisions on drafts, and read and approved the final paper.

## Additional information

**Competing interests:** The authors declare no competing interests.

