## [Peer Review File · Nature Communications]

Reviewers' Comments:

Reviewer #1:

Remarks to the Author:

- What are the major claims of the paper?

The major claims of the paper are that they have identified regulatory genetic variation that is closely associated with alternative social and solitary forms within the sweat bee *Lasioglossum albipes*.

- Are the claims novel? If not, please identify the major papers that compromise novelty

This finding is novel, to my knowledge it is the first study that identifies genetic variation related to solitary vs social forms in any eusocial species.

- Will the paper be of interest to others in the field?

Yes, most certainly! It is a unique species system with great power to study the evolution and genetics of sociality, and the fact that they not only identified genetic variants, but that they were regulatory variants, and their main variant was for a gene with well-established ties to behavior and social behavior in a wide variety of species, make this study especially notable.

- Will the paper influence thinking in the field?

I think the results of this study are not surprising, but rather confirm several long held ideas—regulatory variation in social evolution being important, and the fact that this can be genetically based within a socially polymorphic species. But it is important to get empirical data to test long held assumptions and ideas!

- Are the claims convincing? If not, what further evidence is needed?

The data and analyses are sound and the results quite convincing that they have indeed identified genetic variations that distinguish solitary and social populations of this bee. I appreciate that they put a large amount of effort into improving the genome assembly (the previous assembly was definitely not good enough and they made major improvements), they did extensive sampling with good sample sizes, and they went to the trouble of doing the luciferase reporter assay, which was a very nice addition to more directly address the potential regulatory role of the identified SNPS for syntaxin.

- Are there other experiments that would strengthen the paper further? How much would they improve it, and how difficult are they likely to be?

I think this is a solid publishable unit.

- Are the claims appropriately discussed in the context of previous literature?

Yes, but I have one major point that I think needs to be addressed with the basic premise of the study. Although this socially polymorphic sweat bee system is very interesting, and the presented results quite convincing, one issue that I think the authors need to address is that they are studying the loss of sociality. They are not studying its origins, which is purported to be the big question they address. I think their results are still valuable, and suggest the identified genetic variants might be important in the maintenance of sociality in these bees, but they need to be more clear about how studying a repeated loss of sociality and its genetic basis can provide insights into the evolution of sociality. They basically assume this is obvious throughout but I strongly feel that logic is not obvious. It could be completely different genetic variants contributing to the origins and maintenance of sociality vs its loss. I think that the utility of the system could indeed be convincingly argued for, so I don't think it sinks the paper, but I do think this should to be carefully addressed in a revision, with appropriate citations on the approach and system.

- If the manuscript is unacceptable in its present form, does the study seem sufficiently promising that the authors should be encouraged to consider a resubmission in the future?

Yes

- Is the manuscript clearly written? If not, how could it be made more accessible?

Very well written, very clear and informative figures.

- Could the manuscript be shortened to aid communication of the most important findings?

I think the length is fine.

- Have the authors done themselves justice without overselling their claims?

I think the authors have not oversold their claims, but they need to be more clear and careful (as stated above) that what they have identified are genetic variants for sociality loss, and better justify how this is useful information for understanding eusocial evolution.

- Have they been fair in their treatment of previous literature?

Yes, given length constraints.

- Have they provided sufficient methodological detail that the experiments could be reproduced?

Methods seemed fine given length constraints.

- Is the statistical analysis of the data sound?

I have no issues with the statistics or bioinformatics analysis.

- Should the authors be asked to provide further data or methodological information to help others replicate their work? (Such data might include source code for modelling studies, detailed protocols or mathematical derivations).

The supplements are quite detailed. They deposited data in online archives. Seems fine.

- Are there any special ethical concerns arising from the use of animals or human subjects?

No

Other comments to consider. I first quote the section of the manuscript referred to, and then provide the associated comment below that.

61 analysis (PCA) using a set of LD-pruned SNPs. The results demonstrate that individuals largely
62 cluster by population, but populations do not cluster by social behavior (Figure 2a). A population
63 tree mirrors these relationships, highlighting the repeated clustering of both social and solitary
64 individuals and populations (Figure 2b).

The wording is slightly confusing. The first (and then subsequent) statements says that populations do not cluster by social behavior. The next line says there is clustering of ..."social and solitary ... populations". Maybe clarify to say that some social and solitary populations cluster together, rather than clustering by social type.

I also find it surprising that AUD and VEN cluster together so well, despite the large geographic distance. I think the authors should address this unusual finding, as it could be construed as casting doubt on the validity of their analyses. Any idea what is going on there? Movement along a mountain range, habitat type, or valley?

99 mechanisms shaping facultative social behavior in halictids appear similar to those shaping obligate eusocial behavior in other species 100 .

Needs rewording as this is too strongly stated. This implies causality in both studies, which has not been demonstrated in either. Rather, there are similarities in the putative functions associated with sociality in both scenarios.

102 Of the 194 SNPs associated with social behavior, 21 are clustered in or nearby six candidate 103 genes implicated in human autism

257 . To assess overlap with autism-associated genes in humans, we used a
258 curated set of genes associated with Autism Spectrum Disorder in humans from the Simons 259 Foundation Autism Research Initiative (SFARI).

I realize there are length restrictions, but there should be more mention of how this was determined. The methods state there were BLASTs to human genes, orthology was established, and each of these genes was found in a human study, but how were those GWAS, or how were the connections in

humans established? A bit more is needed here, it is quite vague otherwise and hard to know how much credence to place in the autism connection.

Figure 4c should be 4a as this result is presented first in the text.

159 expression in wild-caught individuals in *L. albipes*. Future transgenic studies in *L. albipes* will 160 help elucidate the precise neurobiological mechanism by which *syx1a* affects the repeated 161 evolution of transitions in social behavior within this single species.

This is quite the teaser, especially as transgenics are not yet developed in this fairly newly developed system (at least not published)! Perhaps state to be more general, saying, future causal studies (this could encompass transgenics, RNAi, pharmacological approaches, etc.)

Final thought-- The syntaxin result is very interesting, as are the connections to behavior in a wide spectrum of other species! The FST and LD results on the nearby SNPs present a very compelling case for something interesting going on in the regulatory region of this gene in social vs solitary forms. Exciting results!

Reviewer #2:

Remarks to the Author:

Review of Nat Comm article "The genetic basis of a social polymorphism in halictid bees" by Kocher et al.

Summary

The authors investigate the genetic basis of social behavior in a bee using sequencing and gene expression analyses. They find genetic variants that are associated with behavior differences between populations. They interpret their results in light of the evolution of social behavior in insects.

Overall, I found this to be an interesting a well-conceived study. The article is well written and clear. The authors find interesting associations between social behavior and SNPs. The work helps us understand more about the genetic basis of social behavior.

The only downside is that the authors can't really pin down the genetic changes that cause variation in sociality. The authors' focus on the gene *syx1a* is interesting and the associated experiments are useful. But the data are not convincing that *syx1a*, and the associated SNPs, is 'the' gene causing differences. The gold standard is an in vivo knock out or gene replacement, which could then show a change in phenotype. Unfortunately, such experiments are effectively impossible in this species. And so the work is still quite interesting in its own right. I have only a few minor comments.

Minor comments

Fig 1 The figure legend explanation of the colors was initially confusing. Why are the populations shown in different shades of blue? Or orange? And what does green color mean? I came to understand in subsequent figures that the colors were meant to differentiate populations. But I was initially confused by this figure.

59 The interpretation of Fst can be complex. Can the authors provide some other estimates from other

insects so this number can be put in context? Also, is the estimate of F_{st} statistically significant? If so, what is the p value?

81 The authors state that “many of these differences are located in potential regulatory regions.” But how exactly are ‘potential regulatory regions’ being defined here? Please be explicit, and cite some appropriate literature to back up your suggestion.

128 The authors focus on brain gene expression in their analyses. There is nothing wrong with this, as brain gene expression is likely important to the function of *syx1a*. But I did wonder if the gene was expressed elsewhere in the body besides the brain. Is it? Did the authors check?

The statement in line 159 is an overreach. Indeed, support that *syx1a* is ‘the’ gene would be much stronger if the authors had conducted their in vitro expression analyses on other genes with SNPs associated with social behavior and found no effects. But the analyses on *syx1a* alone are not particularly convincing.

Reviewer #3:

Remarks to the Author:

The manuscript “The genetic basis of a social polymorphism in halictid bees” is an interesting study identifying genomic differences between three populations of solitary bees and three populations of social bees. This study is of broad interest. The authors do a good job of telling a story emphasizing expression differences that appear to be caused by SNPs regulating their strongest candidate gene, whose orthologs influence social behavior in diverse animals. However, in the process of telling this story, the authors gloss over important details. I strongly recommend that the authors more carefully and thoroughly present their results and explain what their results mean for the genetic basis of social polymorphism in their study species. I have the following major suggestions. Overall, these suggested changes should not be difficult to make, but I think they will greatly improve the manuscript.

1. Heritability of social polymorphism. The authors start their argument by stating that “variation in eusociality ... is fixed among populations and has a strong genetic component (ref3)”. Reference 3 is a common garden study that appears to show in fact that variation in eusociality is not fixed in this species. In any case, the authors should actually estimate the heritability of the social polymorphism (i.e. the proportion of variance in phenotype explained by typed genotypes in GEMMA).

2. Behavior-associated SNPs. The authors found 194 SNPs “associated with social behavior”. However, variation in social behavior is “strongly correlated with season length and mean temperature” (figure 1 caption), so that the identified SNPs may causally influence behavior, some aspect of local adaptation, or may have no phenotypic effect at all. The authors should explicitly state this important caveat. Indeed, line 64-67, they already suggest that repeated shifts in social behavior could be “perhaps as a result of local adaptation”. Similarly, it is imprecise to label alleles that are relatively more or less common in social/solitary populations as a “social allele” or “solitary allele”. These labels should be clearly defined, with the caveat that these alleles are not fixed in the social/solitary populations, and are only putatively associated with the social polymorphism per se. Lines 126-128 should also clarify that the sequenced “social individuals” and “solitary bees” are actually males (and not females that express the social behavior) from populations that contain either social or solitary females. Line 133, the authors stress “exceptionally high levels of F_{ST} (0.38 and 0.37)” for the two SNPs in *syx1a*. These values are certainly very high for randomly chosen genes in the genome in the face of gene flow. But are they high for a gene that putatively underlies the observed social polymorphism? The authors should carefully explain that in fact none of the identified SNPs were consistently fixed between social

and solitary populations, and also briefly explain what this means for the genetic basis of the social polymorphism (i.e. presumably it means that the social polymorphism is a complex trait, perhaps involving a threshold). It would also be very helpful if the authors explicitly stated somewhere the estimated allele frequencies of the 194 SNPs between social and solitary populations (and also for each of the 6 populations).

3. Importance of regulatory versus coding change. The authors stress that many of the identified SNPs occur in regulatory regions, however, I had a difficult time assessing the strength of this argument. How do the definitions used for regulatory regions affect the results? The authors stress that only 17/194 SNPs are predicted to be non-synonymous variants. How does this compare to expectations? At first glance, ~9% non-synonymous variants appears to be much more than expected by chance. Compared to expectation, are there actually more SNPs in assumed regulatory versus coding regions? I strongly recommend that the authors include a table with the 9 genes that are predicted to have nonsynonymous changes. From the supplemental table, it appears that several genes (e.g., *Trap1*, *Cklalpha*, *BicD*) have both missense variants and intron variants with F_{ST} similar to that stressed for *Syx1A*. Why not also at least briefly mention that SNPs in these genes might also be important? It also appears that 11/17 of the missense variants are predicted to fall in genes with no identifiable orthologs. The authors should briefly explain what this result means. Similarly, the GO analysis clearly only considers genes with identifiable orthologs. How many of the 62 candidate genes with coding or putative regulatory changes had identifiable orthologs?

4. Methods. Please explain why a FDR-corrected significance threshold of 5×10^{-5} was used. Please cite references supporting this approach. Line 219, haploid males were not just used "initially" to generate resequencing data, correct? Excluding variants with heterozygous calls in 15/150 individuals does not seem very conservative, in particular given that the individuals are haploid. How does this threshold affect the results? The supplemental tables are not clear. Please provide a clear description of what each column means.

5. References. The references cited were sparse. For example, I was surprised that the authors did not even cite Kapheim et al 2015 "Genomic signatures of evolutionary transitions from solitary to group living", which also stresses the importance for regulatory evolution underlying social complexity. Other empirical papers also stress this, as do theory papers.

Responses to reviewers' comments are in blue.

Reviewer #1 (Remarks to the Author):

- What are the major claims of the paper?

The major claims of the paper are that they have identified regulatory genetic variation that is closely associated with alternative social and solitary forms within the sweat bee *Lasioglossum albipes*.

- Are the claims novel? If not, please identify the major papers that compromise novelty

This finding is novel, to my knowledge it is the first study that identifies genetic variation related to solitary vs social forms in any eusocial species.

- Will the paper be of interest to others in the field?

Yes, most certainly! It is a unique species system with great power to study the evolution and genetics of sociality, and the fact that they not only identified genetic variants, but that they were regulatory variants, and their main variant was for a gene with well-established ties to behavior and social behavior in a wide variety of species, make this study especially notable.

- Will the paper influence thinking in the field?

I think the results of this study are not surprising, but rather confirm several long held ideas— regulatory variation in social evolution being important, and the fact that this can be genetically based within a socially polymorphic species. But it is important to get empirical data to test long held assumptions and ideas!

- Are the claims convincing? If not, what further evidence is needed?

The data and analyses are sound and the results quite convincing that they have indeed identified genetic variations that distinguish solitary and social populations of this bee. I appreciate that they put a large amount of effort into improving the genome assembly (the previous assembly was definitely not good enough and they made major improvements), they did extensive sampling with good sample sizes, and they went to the trouble of doing the luciferase reporter assay, which was a very nice addition to more directly address the potential regulatory role of the identified SNPs for syntaxin.

- Are there other experiments that would strengthen the paper further? How much would they improve it, and how difficult are they likely to be?

I think this is a solid publishable unit.

- Are the claims appropriately discussed in the context of previous literature?

Yes, but I have one major point that I think needs to be addressed with the basic premise of the study. Although this socially polymorphic sweat bee system is very interesting, and the presented results quite convincing, one issue that I think the authors need to address is that they are studying the loss of sociality. They are not studying its origins, which is purported to be the big question they address. I think their results are still valuable, and suggest the identified genetic variants might be important in the maintenance of sociality in these bees, but they need to be more clear about how studying a repeated loss of sociality and its genetic basis can provide insights into the evolution of sociality. They basically assume this is obvious throughout but I strongly feel that logic is not obvious. It could be completely different genetic variants contributing to the origins and maintenance of sociality vs its loss. I think that the utility of the system could indeed be convincingly argued for, so I don't think it sinks the paper, but I do think this should be carefully addressed in a revision, with appropriate citations on the approach and system.

We thank the reviewer for this thoughtful comment. We have modified the text of the manuscript to elaborate on this idea.

Lines 173-179:

“Because the shifts in social behavior in *L. albipes* are more likely to be the product of losses of eusociality rather than independent gains², many of the genetic associations with variation in social behavior identified in this study are likely to represent mechanisms contributing to the maintenance or disruption of eusociality rather than to its precise evolutionary origins. Nonetheless, here the derived allele of *syx1a*, which occurs in an extended haplotype block, is associated with eusociality, suggesting this allele may be contribute to the maintenance or elaboration of social behavior in this system.”

• If the manuscript is unacceptable in its present form, does the study seem sufficiently promising that the authors should be encouraged to consider a resubmission in the future?

Yes

• Is the manuscript clearly written? If not, how could it be made more accessible?

Very well written, very clear and informative figures.

• Could the manuscript be shortened to aid communication of the most important findings?

I think the length is fine.

• Have the authors done themselves justice without overselling their claims?

I think the authors have not oversold their claims, but they need to be more clear and careful (as stated above) that what they have identified are genetic variants for sociality loss, and better justify how this is useful information for understanding eusocial evolution.

• Have they been fair in their treatment of previous literature?

Yes, given length constraints.

• Have they provided sufficient methodological detail that the experiments could be reproduced?

Methods seemed fine given length constraints.

• Is the statistical analysis of the data sound?

I have no issues with the statistics or bioinformatics analysis.

• Should the authors be asked to provide further data or methodological information to help others replicate their work? (Such data might include source code for modelling studies, detailed protocols or mathematical derivations).

The supplements are quite detailed. They deposited data in online archives. Seems fine.

• Are there any special ethical concerns arising from the use of animals or human subjects?

No

Other comments to consider. I first quote the section of the manuscript referred to, and then provide the associated comment below that.

61 analysis (PCA) using a set of LD-pruned SNPs. The results demonstrate that individuals largely
62 cluster by population, but populations do not cluster by social behavior (Figure 2a). A population
63 tree mirrors these relationships, highlighting the repeated clustering of both social and solitary
64 individuals and populations (Figure 2b).

The wording is slightly confusing. The first (and then subsequent) statements says that populations

do not cluster by social behavior. The next line says there is clustering of ...”social and solitary ... populations”. Maybe clarify to say that some social and solitary populations cluster together, rather than clustering by social type.

We have rephrased this sentence to read as follows:

Lines 62-66:

“Next, we implemented a principal component analysis (PCA) using a set of LD-pruned SNPs. The results demonstrate that individuals largely cluster by population, but populations do not cluster by social behavior (Figure 2a). A population tree mirrors these relationships, with social and solitary populations repeatedly clustering together rather than separating by social form (Figure 2b).”

I also find it surprising that AUD and VEN cluster together so well, despite the large geographic distance. I think the authors should address this unusual finding, as it could be construed as casting doubt on the validity of their analyses. Any idea what is going on there? Movement along a mountain range, habitat type, or valley?

Because we compare individuals from all social versus all solitary populations in the genome-wide association (i.e. we pool all social individuals and all solitary individuals and use a genetic relatedness matrix as a co-variate in GEMMA), this observation does not impact the results or interpretation of the work presented in this manuscript.

However, we also find this result surprising, and have spent a substantial amount of effort attempting to uncover technical errors that could be the source of this pattern. Samples for each of these populations were collected across multiple sampling events and field seasons, and all samples were processed in random order for visual identification, DNA barcoding, DNA extraction and library preparation. We have also replicated this pattern across independently collected and sequenced samples at two different sequencing facilities, therefore, this grouping is unlikely to be due to an error in sample labeling or a similar technical error. Furthermore, the pattern holds regardless of analytical method (in the paper, this is demonstrated with the genetic PCA and with the population tree constructed using fourfold degenerate sites across the genome).

At this point, we are not sure what exactly is driving this pattern, and a current project involves sampling *L. albipes* across a broader geographic range and conducting detailed demographic and biogeographic analyses to try to better understand what factors could underlie this result. However, this work is ongoing and, we feel, beyond the scope of this initial population genetic study.

99 mechanisms shaping facultative social behavior in halictids appear similar to those shaping obligate eusocial behavior in other species 100 .

Needs rewording as this is too strongly stated. This implies causality in both studies, which has not been demonstrated in either. Rather, there are similarities in the putative functions associated with sociality in both scenarios.

We have modified the text to read “mechanisms linked to facultative social behavior in halictids appear similar to those associated with obligate eusocial behavior in other species” (lines 130-131).

102 Of the 194 SNPs associated with social behavior, 21 are clustered in or nearby six candidate
103 genes implicated in human autism

257 To assess overlap with autism-associated genes in humans, we used a

258 curated set of genes associated with Autism Spectrum Disorder in humans from the Simons 259 Foundation Autism Research Initiative (SFARI).

I realize there are length restrictions, but there should be more mention of how this was determined. The methods state there were BLASTs to human genes, orthology was established, and each of these genes was found in a human study, but how were those GWAS, or how were the connections in humans established? A bit more is needed here, it is quite vague otherwise and hard to know how much credence to place in the autism connection.

We have included more details on these methods and the SFARI gene set in the main text.

Lines 277-284:

“To assess overlap with autism-associated genes in humans, we used a curated set of genes associated with Autism Spectrum Disorder from the Simons Foundation Autism Research Initiative (SFARI). This gene list includes all known human genes associated with ASD, including genes identified in genome-wide association studies, genes where rare mutations have been linked to ASD, and genes linked to syndromic autism. We used the full list of genes included in this database. We matched *L. albipes* genes to human orthologs using of reciprocal best-blast hits between *L. albipes* and human, and also confirmed annotations with SwissProt homologies generated using the trinotate pipeline on the Lalb_v3 OGS⁴⁰.”

Figure 4c should be 4a as this result is presented first in the text.

Thank you. This figure has been updated and references to each panel have also been updated in the text.

159 expression in wild-caught individuals in *L. albipes*. Future transgenic studies in *L. albipes* will
160 help elucidate the precise neurobiological mechanism by which *syx1a* affects the repeated 161
evolution of transitions in social behavior within this single species.

This is quite the teaser, especially as transgenics are not yet developed in this fairly newly developed system (at least not published)! Perhaps state to be more general, saying, future causal studies (this could encompass transgenics, RNAi, pharmacological approaches, etc.)

We have modified the text to read (lines 169-171):

“Future functional studies in *L. albipes* will help elucidate the precise neurobiological mechanisms by which *syx1a* and other associated genes shape variation in social behavior within this single species.”

Final thought-- The syntaxin result is very interesting, as are the connections to behavior in a wide spectrum of other species! The FST and LD results on the nearby SNPs present a very compelling case for something interesting going on in the regulatory region of this gene in social vs solitary forms. Exciting results!

Thank you for your enthusiasm and advice!

Reviewer #2 (Remarks to the Author):

Review of Nat Comm article “The genetic basis of a social polymorphism in halictid bees” by Kocher et al.

Summary

The authors investigate the genetic basis of social behavior in a bee using sequencing and gene expression analyses. They find genetic variants that are associated with behavior differences between populations. They interpret their results in light of the evolution of social behavior in insects.

Overall, I found this to be an interesting a well-conceived study. The article is well written and clear. The authors find interesting associations between social behavior and SNPs. The work helps us understand more about the genetic basis of social behavior.

The only downside is that the authors can't really pin down the genetic changes that cause variation in sociality. The authors' focus on the gene *syx1a* is interesting and the associated experiments are useful. But the data are not convincing that *syx1a*, and the associated SNPs, is 'the' gene causing differences. The gold standard is an in vivo knock out or gene replacement, which could then show a change in phenotype. Unfortunately, such experiments are effectively impossible in this species. And so the work is still quite interesting in its own right. I have only a few minor comments.

Minor comments

Fig 1 The figure legend explanation of the colors was initially confusing. Why are the populations shown in different shades of blue? Or orange? And what does green color mean? I came to understand in subsequent figures that the colors were meant to differentiate populations. But I was initially confused by this figure.

We use different shades of blues and greens to denote social populations of this species, and different shades of red and orange to denote solitary populations. We feel the different shades are important to help readers distinguish among different populations of each social form. We have attempted to better clarify this in the legends for figures 1 and 2:

Figure 1 legend:

“Populations in western France are eusocial (blue and green colors), while populations in the east are solitary (red and orange colors).”

Figure 2 legend:

“Genetic principal component analysis of 6 populations: 3 social (blues and greens) and 3 solitary (reds and oranges).”

59 The interpretation of F_{ST} can be complex. Can the authors provide some other estimates from other insects so this number can be put in context? Also, is the estimate of F_{ST} statistically significant? If so, what is the p value?

The F_{ST} estimates for *L. albipes* are similar to estimates comparing honey bee populations, where F_{ST} ranges from 0.05-0.2 across different populations from the same mitochondrial groups (Wallberg *et al.* 2014). These values are also on par with observations from several ant species, where F_{ST} has been estimated between 0.0-0.26 depending on the species (Sanetra & Crozier 2003; Sundström *et al.* 2005; Doums *et al.* 2008).

We have now included permutation tests to assess the significance of the F_{ST} between social and solitary populations. Social designations were randomly shuffled 1000 times for each individual and F_{ST} was estimated as previously described using a randomly-selected subset of 100,000 SNPs. None of the permutations produced an $F_{ST} \geq 0.06$ (mean permuted $F_{ST} = 0.0009 \pm 0.0019$), suggesting that the differentiation observed among social and solitary forms is significantly more than expected by chance ($p < 0.001$).

We have included this in the main text, lines 58-62:

“The mean F_{ST} between social and solitary forms is 0.06 (mean permuted $F_{ST} = 0.0009 \pm 0.0019$; $p < 0.001$), suggesting that there is, or recently has been, gene flow among behavioral groups. These estimates are similar to those from honey bee populations, where F_{ST} ranges from 0.05-0.2 across different populations⁵.”

Lines 254-257:

“To estimate the statistical significance of differentiation between social and solitary populations, social designations were randomly shuffled 1000 times for each individual, and F_{ST} was estimated using a randomly-selected subset of 100,000 SNPs.”

81 The authors state that “many of these differences are located in potential regulatory regions.” But how exactly are ‘potential regulatory regions’ being defined here? Please be explicit, and cite some appropriate literature to back up your suggestion.

Putative regulatory regions included: 5kb upstream of TSS and 1kb downstream of the last codon. These classifications are based on (McLean *et al.* 2010). We have now included this citation in the main text.

Lines 88-92:

“The candidate SNPs fall within 10kb of 62 genes, and many of these differences are located in potential regulatory regions¹⁰. In fact, 40% of identified SNPs fall nearby genes, either 5kb upstream of the transcription start site ($n=45$) or 1kb downstream of the last codon ($n=32$), a significantly greater proportion of variants in these regions than expected by chance (hypergeometric test, $p = 1.4 \times 10^{-5}$).”

128 The authors focus on brain gene expression in their analyses. There is nothing wrong with this, as brain gene expression is likely important to the function of *syx1a*. But I did wonder if the gene was expressed elsewhere in the body besides the brain. Is it? Did the authors check?

We only used dissected brain tissue to generate the qPCR results presented in this study (Figure 4b). We focused on brain tissue because, as the reviewer has pointed out, behaviors should be tightly linked to brain gene expression patterns. That said, *syx1a* is expressed throughout development and in multiple tissues (based on expression data in *Drosophila*), and as we continue to investigate the functional link between *syx1a* expression levels and social behavior, we will examine multiple developmental timepoints and tissues.

The statement in line 159 is an overreach. Indeed, support that *syx1a* is ‘the’ gene would be much stronger if the authors had conducted their in vitro expression analyses on other genes with SNPs associated with social behavior and found no effects. But the analyses on *syx1a* alone are not particularly convincing.

We think the reviewer is referring to this statement: “Future transgenic studies in *L. albipes* will help elucidate the precise neurobiological mechanism by which *syx1a* affects the repeated evolution of transitions in social behavior within this single species”. We have modified this text to read:

Lines 169-171:

“Future functional studies in *L. albipes* will help elucidate the precise neurobiological mechanisms by which *syx1a* and other associated genes shape variation in social behavior within this single species.”

We thank the reviewer for pointing this out and hope that the text now makes it clear that we do not believe that *syx1a* is the only gene shaping variation in social behavior in this species. Social behavior is a complex trait involving many aspects of behavior and physiology. In this study, we have identified several regions in the genome that are associated with variation in sociality in *L. albipes*. Because functional tests on all of the variants identified in this study is not feasible, we have focused on validation of a single candidate gene, *syx1a*, because it showed the strongest and most consistent association with eusociality in our analyses.

Reviewer #3 (Remarks to the Author):

The manuscript “The genetic basis of a social polymorphism in halictid bees” is an interesting study identifying genomic differences between three populations of solitary bees and three populations of social bees. This study is of broad interest. The authors do a good job of telling a story emphasizing expression differences that appear to be caused by SNPs regulating their strongest candidate gene, whose orthologs influence social behavior in diverse animals. However, in the process of telling this story, the authors gloss over important details. I strongly recommend that the authors more carefully and thoroughly present their results and explain what their results mean for the genetic basis of social polymorphism in their study species. I have the following major suggestions. Overall, these suggested changes should not be difficult to make, but I think they will greatly improve the manuscript.

1. Heritability of social polymorphism. The authors start their argument by stating that “variation in eusociality ... is fixed among populations and has a strong genetic component (ref3)”. Reference 3 is a common garden study that appears to show in fact that variation in eusociality is not fixed in this species.

We respectfully disagree with the reviewer’s interpretation of the Plateaux Quenu study (reference #3 in this manuscript). In this study, females from eusocial and non-eusocial populations were reared in a common laboratory environment under photoperiods associated with eusocial conditions (long days) and solitary conditions (short days). Under both conditions, all eusocial foundresses (n=43) produced workers, and interestingly, even when the first set of workers was removed from eusocial nests, all eusocial foundresses produced replacement workers before producing a future reproductive generation.

We think the variation the reviewer is referring to is the finding that out of 22 solitary foundresses reared in the common garden, 2 females produced a single worker each. Both of these females were collected from the same population (in Nancy, France) where climatic conditions are warmer on average than any of the other studied solitary populations. Because only 2/22 solitary foundresses produced a single worker, and because these 2 females both came from the same (and warmest) solitary population, we interpret this result as one that demonstrates that the variation in social behavior in *L. albipes* occurs across populations and is likely to have a strong genetic component. This is a viewpoint shared by Dr. Plateaux Quenu, who gave us background material, advised us, and showed us her original collecting sites and protocols when we began this research. Finally, we also note that our study did not include samples from the population in Nancy because, despite numerous collecting attempts over several years, we were unable to find bees at this site.

Nonetheless, we thank the reviewer for pointing out that the behavior is not fixed across all populations, and we have rephrased this sentence to read:

Lines 36-38:

“in *L. albipes* differences in social behavior primarily occur among populations and common garden experiments³ suggest there is a strong genetic component underlying this variation.”

In any case, the authors should actually estimate the heritability of the social polymorphism (i.e. the proportion of variance in phenotype explained by typed genotypes in GEMMA).

This is a question to which we would love to have a quantitative answer. but unfortunately, it's a difficult question to address given our data. There are a variety of approaches to estimate heritability, from parent-offspring regressions to pedigrees, to (more recently) the use of REML with random population samples. However, for REML estimates to be meaningful, sample sizes need to be much larger than what we currently have (Visscher *et al.* 2014). We can estimate the proportion of variance in phenotype explained by the typed genotypes in GEMMA, but this model is in many ways over-fitted (in our case, the GEMMA estimate is close to 0.99). This is not surprising given that we are fitting a linear model using >2 million SNPs, and it likely represents an overestimate of the genetic component of this trait.

Other approaches that might be better suited to estimating heritability in this study include GCTA (Yang *et al.* 2011), which is an approach specifically designed to estimate the genetic contribution to a phenotype using SNPs. GCTA uses all genotyped polymorphisms (or a large subset) to estimate a genetic relationship matrix (GRM) and establish the degree of shared genotypes between pairs of individuals in a sample. Using restricted maximum likelihood (REML) and considering SNPs as random effects, GCTA estimates the variance explained by all SNPs given the GRM. In other words, GCTA aims to explain the degree of phenotypic resemblance between individuals in the sample by the proportion of alleles they share. The problem is that GCTA also requires a large number of individuals (on the order of thousands rather than hundreds; Visscher *et al.* 2014) to properly estimate the variance. Thus, the most meaningful estimate we can provide with our data focuses on our significant SNPs, where the proportion of genetic variance explained for this trait is 0.7003 (95% CI = 0.52, 0.88) indicating that at least ~70% of the total behavioral variation observed in our samples can be attributed to those genetic factors.

We are starting to rear these bees in the lab ourselves, and in the future, we will be better able to address this question using more traditional quantitative-genetics approaches based on classic breeding designs.

2. Behavior-associated SNPs. The authors found 194 SNPs “associated with social behavior”. However, variation in social behavior is “strongly correlated with season length and mean temperature” (figure 1 caption), so that the identified SNPs may causally influence behavior, some aspect of local adaptation, or may have no phenotypic effect at all. The authors should explicitly state this important caveat. Indeed, line 64-67, they already suggest that repeated shifts in social behavior could be “perhaps as a result of local adaptation”.

We have expanded on this in lines 105-108:

“These gene functions suggest a link to behavioral variation, but because behavioral variation is tightly linked to environmental conditions in this species, some may also reflect associations driven by differences in climatic conditions.”

Similarly, it is imprecise to label alleles that are relatively more or less common in social/solitary populations as a “social allele” or “solitary allele”. These labels should be clearly defined, with the caveat that these alleles are not fixed in the social/solitary populations, and are only putatively associated with the social polymorphism per se.

We agree that this is not a perfect label, but we think it's important to note which behavioral form each allele was associated with, particularly given that the phenotypic contrasts made in this association were also "social" vs "solitary".

We have added more precise language about this in the main text, lines 154-156:

"either the allele at higher frequency in social populations (henceforth the "social allele") or the allele at higher frequency in solitary forms (the "solitary allele")"

Lines 126-128 should also clarify that the sequenced "social individuals" and "solitary bees" are actually males (and not females that express the social behavior) from populations that contain either social or solitary females.

This has been specified in lines 296-297 in the methods:

"We collected five social and five solitary individuals (males) from two sites: RIM (social) and BRS (solitary), as previously described."

We have also edited the main text to better clarify (lines 135-137):

"we used quantitative, reverse transcription PCR and found that bees from social populations have significantly higher levels of *syx1a* brain gene expression than those from solitary populations"

Line 133, the authors stress "exceptionally high levels of F_{ST} (0.38 and 0.37)" for the two SNPs in *syx1a*. These values are certainly very high for randomly chosen genes in the genome in the face of gene flow. But are they high for a gene that putatively underlies the observed social polymorphism?

This is a difficult question to answer because we do not believe that the SNPs in *syx1a* are the only variants underlying the social polymorphism in this species. As we mention above, social behavior is a clearly a complex trait involving many aspects of behavior and physiology. We have focused on validation of a single candidate gene, *syx1a*, because it showed the strongest and most consistent association with eusociality in our analyses, but we cannot easily compare it with other genes underlying the social polymorphism because these genes may very well act in concert with each other.

The authors should carefully explain that in fact none of the identified SNPs were consistently fixed between social and solitary populations, and also briefly explain what this means for the genetic basis of the social polymorphism (i.e. presumably it means that the social polymorphism is a complex trait, perhaps involving a threshold).

We have added the following text (lines 79-86):

"First, no variant was consistently fixed among all social versus all solitary populations, suggesting that there is not a single, shared locus shaping variation in social behavior in this species but rather that the genetic underpinnings of this trait are complex. Concordantly, we found 8 regions containing 194 SNPs passing a genome-wide, FDR-corrected significance threshold⁹ of 5×10^{-5} (which roughly corresponds to a raw-p threshold of 5×10^{-9} ; Figure 3; Extended Data Table 3), suggesting that multiple regions throughout the genome contribute to intraspecific behavioral variation in *L. albipes*."

It would also be very helpful if the authors explicitly stated somewhere the estimated allele frequencies of the 194 SNPs between social and solitary populations (and also for each of the 6 populations).

Allele frequencies for each SNP have now been added to the supplementary table. Because all social individuals and solitary individuals have been pooled across populations for the statistical analyses (and population-level variation has been accounted for as a covariate in these models), we

have accordingly reported the estimated allele frequencies for social and solitary forms.

3. Importance of regulatory versus coding change. The authors stress that many of the identified SNPs occur in regulatory regions, however, I had a difficult time assessing the strength of this argument. How do the definitions used for regulatory regions affect the results? The authors stress that only 17/194 SNPs are predicted to be non-synonymous variants. How does this compare to expectations? At first glance, ~9% non-synonymous variants appears to be much more than expected by chance. Compared to expectation, are there actually more SNPs in assumed regulatory versus coding regions?

I strongly recommend that the authors include a table with the 9 genes that are predicted to have nonsynonymous changes. From the supplemental table, it appears that several genes (e.g., Trap1, Cklalpha, BicD) have both missense variants and intron variants with F_{ST} similar to that stressed for Syx1A. Why not also at least briefly mention that SNPs in these genes might also be important?

We have now included these mutations in a separate, supplemental table as requested (Table S8).

We have also modified the text to include discussion of these candidates and the potential importance of coding sequence changes.

Lines 92-94:

“Moreover, 17 of these 194 SNPs, located in 9 different genes, are predicted to be non-synonymous variants (Extended Data Table 4; hypergeometric test, $p=1.3 \times 10^{-11}$), and 8 variants occur at synonymous sites (hypergeometric test, $p=0.02$).”

Lines 96-98:

“Taken together, these results suggest that changes in both gene regulation and coding sequence play an important role in the social polymorphism within this species.”

It also appears that 11/17 of the missense variants are predicted to fall in genes with no identifiable orthologs. The authors should briefly explain what this result means. Similarly, the GO analysis clearly only considers genes with identifiable orthologs. How many of the 62 candidate genes with coding or putative regulatory changes had identifiable orthologs?

We thank the reviewer for pointing this out; this was previously unclear in the methods and supplement. The final columns of annotations presented in the supplemental tables were based on the reciprocal blast transfers of gene annotations from the previous *L. albipes* genome assembly. However, genes that were missed in the initial assembly but annotated in the improved version did not receive annotations with this method, and for this reason, we did not use those annotations for the GO analyses. For the analyses in this paper, we relied on annotations generated by trinotate (Haas *et al.* 2013). We apologize for the confusion these extra columns created; we have now removed them from the supplemental tables.

As should now be clear in the supplemental tables, 29/194 of the significant SNPs associated with 20/62 (32.2%) of candidate genes had no clearly identifiable ortholog or gene ontology assignment. This is essentially the same as background, where ~32.3% of predicted genes in the Lab_v3 OGS did not have identifiable orthologs or GO terms. Regarding the missense variants, only 3/17 did not have an identifiable ortholog in trinotate (17.6%; this number is not significantly higher or lower than expected by chance; $p=0.15$ for the lower tail and $p=0.85$ for upper tail of a hypergeometric distribution).

4. Methods. Please explain why a FDR-corrected significance threshold of 5×10^{-5} was used. Please cite references supporting this approach.

Significance thresholds are infamously arbitrary for genome-wide association studies. Our reported p-values are FDR-corrected, and thus our threshold corresponds to a raw p-value of 5×10^{-9} , which is an order of magnitude more stringent than typical human GWAS, where the significance threshold frequently used in human studies is 5×10^{-8} on raw p-values. A recent paper that reviewed the appropriateness of this cutoff found that it was still appropriate when using a MAF cutoff < 0.05 as we have implemented in this study (Fadista *et al.* 2016).

We have added some clarification on this point in the main text, lines 82-86:

“Concordantly, we found 8 regions containing 194 SNPs passing a genome-wide, FDR-corrected significance threshold⁹ of 5×10^{-5} (which roughly corresponds to a raw-p threshold of 5×10^{-9} ; Figure 3; Extended Data Table 3), suggesting that multiple regions throughout the genome contribute to intraspecific behavioral variation in *L. albipes*.”

Line 219, haploid males were not just used “initially” to generate resequencing data, correct? Excluding variants with heterozygous calls in 15/150 individuals does not seem very conservative, in particular given that the individuals are haploid. How does this threshold affect the results?

Thank you. We have deleted “initially” (now on line 284).

Most heterozygous calls were found in a handful of single libraries with poor quality sequencing (these individuals were filtered out in a subsequent step), and thus the selection of 5, 10, or 15/150 individuals did not greatly impact our results. Furthermore, none of the SNPs that passed our significance threshold included any heterozygous calls.

The supplemental tables are not clear. Please provide a clear description of what each column means.

We have updated the legends for the supplemental tables to include these details.

5. References. The references cited were sparse. For example, I was surprised that the authors did not even cite Kapheim *et al.* 2015 “Genomic signatures of evolutionary transitions from solitary to group living”, which also stresses the importance for regulatory evolution underlying social complexity. Other empirical papers also stress this, as do theory papers.

We thank the reviewer for pointing out this oversight. We have now cited the Kapheim paper in the main text.

References cited:

- Doums C, Cabrera H, Peeters C (2008) Population genetic structure and male-biased dispersal in the queenless ant *Diacamma cyaneiventris*. *Molecular Ecology*, **11**, 2251–2264.
- Fadista J, Manning AK, Florez JC, Groop L (2016) The (in)famous GWAS P-value threshold revisited and updated for low-frequency variants. *European Journal Of Human Genetics*, **24**, 1202–1205.
- Haas BJ, Papanicolaou A, Yassour M *et al.* (2013) De novo transcript sequence reconstruction from RNA-seq using the Trinity platform for reference generation and analysis. *Nature Protocols*, **8**, 1494–1512.
- McLean CY, Bristor D, Hiller M *et al.* (2010) GREAT improves functional interpretation of cis-regulatory regions. *Nat Biotech*, **28**, 495–501.

- Sanetra M, Crozier RH (2003) Patterns of population subdivision and gene flow in the ant *Nothomyrmecia macrops* reflected in microsatellite and mitochondrial DNA markers. *Molecular Ecology*, **12**, 2281–2295.
- Sundström L, Seppä P, Pamilo P (2005) Genetic population structure and dispersal patterns in *Formica* ants – a review. *Annales Zoologici Fennici*, **42**, 163–177.
- Visscher PM, Hemani G, Vinkhuyzen AAE *et al.* (2014) Statistical Power to Detect Genetic (Co)Variance of Complex Traits Using SNP Data in Unrelated Samples (GS Barsh, Ed.). *PLoS Genetics*, **10**, e1004269.
- Wallberg A, Han F, Wellhagen G *et al.* (2014) A worldwide survey of genome sequence variation provides insight into the evolutionary history of the honeybee *Apis mellifera*. *Nature Genetics*, **46**, 1081–1088.
- Yang J, Lee SH, Goddard ME, Visscher PM (2011) GCTA: A Tool for Genome-wide Complex Trait Analysis. *American Journal of Human Genetics*, **88**, 76–82.

Reviewers' Comments:

Reviewer #1:

Remarks to the Author:

Great job addressing my comments!

Reviewer #3:

Remarks to the Author:

I think that the revised manuscript is much improved. I am satisfied with the authors' responses and revisions.